# Universal Value Density Estimation for Imitation Learning and Goal-Conditioned Reinforcement Learning

## Abstract

This work considers two distinct settings: imitation learning and goal-conditioned reinforcement learning. In either case, effective solutions require the agent to reliably reach a specified state (a goal), or set of states (a demonstration). Drawing a connection between probabilistic long-term dynamics and the desired value function, this work introduces an approach that utilizes recent advances in density estimation to effectively learn to reach a given state. We develop a unified view on the two settings and show that the approach can be applied to both. In goal-conditioned reinforcement learning, we show it to circumvent the problem of sparse rewards while addressing hindsight bias in stochastic domains. In imitation learning, we show that the approach can learn from extremely sparse amounts of expert data and achieves state-of-the-art results on a common benchmark.

## 1 Introduction

Effective imitation learning relies on information encoded in the demonstration states. In the past, successful and sample-efficient approaches have attempted to match the distribution of demonstrated states (Ziebart et al., 2008; Ho and Ermon, 2016; Schroecker et al., 2019), reach any state that is part of the demonstrations (Wang et al., 2019; Reddy et al., 2019), or track a reference trajectory to reproduce a specific sequence of states (Peng et al., 2018; Aytar et al., 2018; Pathak et al., 2018). Attempting to reproduce demonstrated states directly allows the agent to exploit structure induced by environment dynamics and to accurately reproduce expert behavior with only a very small number of demonstrations. Commonly, this is framed as a measure to avoid covariate shift (Ross et al., 2011) but the efficacy of such methods on even sub-sampled trajectories (e.g. Ho and Ermon, 2016) and their ability to learn from observation alone (e.g. Torabi et al., 2018; Schroecker et al., 2019) show benefits beyond the avoidance of accumulating errors. Despite significant progress in the field, reproducing a set of demonstrated states efficiently, accurately and robustly remains an open field of research. To address the problem, we may first ask the question of how to reach arbitrary states, a question that has thus far been considered separately in the field of goal-conditioned reinforcement learning. In this paper, we introduce a unified view to goal-conditioned reinforcement learning and imitation learning. We will first address the question of how to reach single states in the former setting and then extend this notion to an imitation learning approach and reproduce distributions of state-action pairs.

Despite significant achievements in the field (Schaul et al., 2015; Andrychowicz et al., 2017; Nair et al., 2018; Sahni et al., 2019), learning to achieve arbitrary goals remains an extremely difficult challenge. In the absence of a suitably shaped reward function, the signal given to the agent can be as sparse as a constant reward if the goal is achieved and 0 otherwise. Hindsight Experience Replay (HER) (Andrychowicz et al., 2017) constitutes an effective and popular heuristic to address this problem; however, the formulation is ad-hoc and does not lend itself readily to the probabilistic reasoning required in a distribution-matching imitation learning approach. Furthermore, HER can be shown to suffer from bias in stochastic domains or when applied to some actor-critic algorithms as we will discuss in Section 3.1. To address this challenge, we introduce Universal Value Density Estimation (UVD). By considering an important subset of goal-conditioned value functions similar to similar to Andrychowicz et al. (2017), namely those corresponding to reward functions that have an optimal agent reach a specific state, we can observe that the value of a state conditioned on a goal is equivalent to the likelihood of the agent reaching the goal from that state. We use normalizing flows

(see Sec. 2.4) to estimate this likelihood from roll-outs and thereby obtain an estimate of the value function. As we will show in Section 5.1, density estimation does not sample goals independently at random and therefore provides a dense learning signal to the agent where temporal-difference learning alone fails due to sparse rewards. This allows us to match the performance of HER in deterministic domains while avoiding bias in stochastic domains and while providing the foundation for our imitation learning approach. We will introduce this approach in Section 3.4 and show that it performs as well as the state-of-the-art on the most common benchmark tasks while significantly outperforming this baseline on a simple stochastic variation of the same domain (Sec. 5.1).

Returning to the imitation learning setting, we propose to extend UVD to match a distribution of expert states and introduce Value Density Imitation Learning (VDI). Our goal is to design an imitation learning algorithm that is able to learn from minimal amounts of expert data using self-supervised environment interactions only. Like prior methods (see Sec. 2.3), VDI's objective is to match the expert's state-action distribution. We achieve this by sampling expert states that the agent is currently not likely to visit and using a goal-conditioned value-function to guide the agent towards those states. We will show in Sec. 4 that this minimizes the KL divergence between the expert's and the agent's state-action distributions and therefore provides an intuitive and principled imitation learning approach. Note that unlike most prior method, expert demonstrations are used in VDI to generate goals rather than to train an intermediate network such as a discriminator or reward function. The value function and density estimate are trained using self-supervised roll-outs alone. This makes intermediate networks much less prone to overfitting and we show in Sec. 5.2 that VDI uses demonstrations significantly more efficiently than the current state-of-the-art in common benchmarks.

## 2 BACKGROUND

### 2.1 MARKOV DECISION PROCESSES

Here, we briefly lay out notation while referring the reader to Puterman (2014) for a detailed review. Markov Decision Processes (MDPs) define a set of states $\mathbb{S}$, a set of actions $\mathbb{A}$, a distribution of initial states $d_0(s)$, Markovian transition dynamics defining the probability (density) of transitioning from state $s$ to $s'$ when taking action $a$ as $p(s'|s, a)$, and a reward function $r(s, a)$. In reinforcement learning, we usually try to find a parametric stationary policy $\mu_\theta : \mathbb{S} \to \mathbb{A}$.[1] An optimal policy maximizes the long-term discounted reward $J_\gamma^r(\theta) = E\left[\sum_{t=0}^\infty \gamma^t r(s_t, a_t)|s_0 \sim d_0, \mu_\theta\right]$ given a discount factor $\gamma$ or, sometimes, the average reward $J^r(\theta) = \int d^{\pi_\theta}(s) r(s, \mu_\theta(s)) ds, a$, where the stationary state distribution $d^\mu(s)$ and the stationary state-action distribution $\rho^\mu(s, a)$ are uniquely induced by $\mu$ under mild ergodicity assumptions. A useful concept to this end is the value function $V^\mu(s) = E\left[\sum_{t=0}^\infty \gamma^t r(s_t, a_t)|s_0 = s, \mu\right]$ or Q function $Q^\mu(s, a) = E\left[\sum_{t=0}^\infty \gamma^t r(s_t, a_t)|s_0 = s, a_0 = a, \mu\right]$ which can be used to estimate the policy gradient $\nabla_\theta J_\gamma(\theta)$ (e.g. Sutton et al., 1999; Silver et al., 2014). Finally, we define as $p_\mu(s, a \xrightarrow{t} s')$ the probability of transitioning from state $s$ to $s'$ after $t$ steps when taking action $a$ in state $s$ and following policy $\mu$ afterwards.

### 2.2 GOAL-CONDITIONED REINFORCEMENT LEARNING

Goal-conditioned Reinforcement Learning aims to teach an agent to solve multiple variations of a task, identified by a goal vector $g$ and the corresponding reward function $r^g(s, a)$. Solving each variation requires the agent to learn a goal-conditioned policy, which we write as $\mu_\theta^g(s)$. Here, we condition the reward and policy explicitly to emphasize that they can be conditioned on different goals. The goal-conditioned value function (GCVF) in this setting is then given by $Q_{r^g}^{\mu^{\bar{g}}}(s, a) = E\left[\sum_{t=0}^\infty \gamma^t r^g(s_t, a_t)|s_0 = s, a_0 = a, \mu^{\bar{g}}\right]$.

To solve such tasks, Schaul et al. (2015) introduce the concept of a Universal Value Function Approximator (UVFA), a learned model $Q_\omega(s, a; g)$ approximating $Q_{r^g}^{\mu^g}(s, a)$, i.e. all value functions where the policy and reward are conditioned on the same goal. We extend this notion to models of value-functions which use a goal-conditioned reward with a single fixed policy. To distinguish such models visually, we write $\tilde{Q}_\omega(s, a; g)$ to refer to models which approximate $Q_{r^g}^\mu(s, a)$ for a given $\mu$.

---

[1]We write the policy as a deterministic function, but all findings hold for stochastic policies as well.

Where $Q_\omega$ represents how good the agent is at achieving any goal if it tries to achieve it, $\tilde{Q}_\omega$ models how good a specific policy is, if the task were to achieve the given goal.

Schaul et al. (2015) show that UVFAs can be trained via temporal difference learning with randomly sampled goals; however, if the reward function is sparse (a common case is an indicator function that tells the agent whether a goal has been achieved), the agent rarely observes a non-zero reward. Hindsight updates (HER) (Andrychowicz et al., 2017) are a straight-forward solution, changing goals recorded in a replay memory based on what the agent has actually achieved in hindsight. This approach is intuitive and can also be extended to work with visual observations (Nair et al., 2018; Sahni et al., 2019); however, it does not yield correct results in stochastic domains. This has previously been noted by Lanka and Wu (2018), who propose a heuristic method to address this issue, but no principled method to use hindsight samples in a completely unbiased way has been proposed to date. Temporal Difference Models (TDMs) (Pong et al., 2018; Nasiriany et al., 2019), a method related to ours, use a squared error function as a terminal value and bootstrapping to back up this value over a finite horizon. Our methods learns to model arbitrary terminal values and allows for bootstrapping over an infinite horizon. TDMs are orthogonal to the concept of hindsight samples.

## 2.3 IMITATION LEARNING

Imitation learning (IL) teaches agents to act given demonstrated example behavior. While the MDP formalism can still be used in this scenario, we no longer have a pre-defined reward function specifying the objective. Instead, we are given a sequence of expert state-action pairs. The goal of imitation learning is to learn a policy $\mu_\theta$ that is equivalent to the expert's policy $\mu^*$ which generated the demonstrated states and actions. This problem is underspecified and different formalizations have been proposed to achieve this goal. The simplest solution to IL is known as Behavioral Cloning (BC) (Pomerleau, 1989) and treats the problem as a supervised learning problem. Using demonstrated states as sample inputs and demonstrated actions as target outputs, a policy can be trained easily without requiring further knowledge of, or interaction with, the environment. While this approach can work remarkably well, it is known to be suboptimal as a standard assumption of supervised learning is violated: predictions made by the agents affect future inputs to the policy (Ross et al., 2011). By learning about the environment dynamics and explicitly trying to reproduce future demonstrated states, the agent is able to learn more robust policies from small amounts of demonstration data. A common and effective formalism is to reason explicitly about matching the distribution of state-action pairs that the agent will see to that of the expert. By learning from interaction with the environment and employing sequential reasoning, approaches such as maximum entropy IRL (Ziebart et al., 2008), adversarial IRL (Fu et al., 2018), GPRIL (Schroecker et al., 2019) or Generative Adversarial Imitation Learning (GAIL) (Ho and Ermon, 2016) lead the agent to reproduce the expert's observations as well as the expert's actions. Especially adversarial approaches (Ho and Ermon, 2016; Fu et al., 2018; Kostrikov et al., 2019) have gained popularity in recent years and train a discriminator to serve as a distance function between the agent's and the expert's state-action distribution. Here, we instead propose to train a generative model to predict multiple time-steps ahead in order to reproduce demonstration states. GPRIL (Schroecker et al., 2019) similarly utilizes a long-term generative model for imitation learning; however, in contrast to VDI, it cannot easily be combined with temporal-difference learning techniques and struggles with larger time-horizons and performs badly on locomotion benchmark tasks. In Section 5.2, we show that VDI cannot only be applied in such domains, but that it outperforms the current state-of-the-art on common benchmark tasks.

## 2.4 NORMALIZING FLOWS

Normalizing flows (van den Oord et al., 2017; Dinh et al., 2016) are capable of learning explicit representations of complex, highly non-linear density functions. In this work, we utilize a simplified version of RealNVP (Dinh et al., 2016) to represent value functions as density functions. A RealNVP model consist of a chain of learned, invertible transformations which transform samples $z \in \mathbb{R}^\mathbb{N}$ from one distribution $p_z$ to samples $x \in \mathbb{R}^\mathbb{N}$ from another distribution $p_x$. The transformations are affine where half of the input features are scaled and shifted by parameters predicted based on the other half which allows for tractable computation of the gradient. The model can therefore be trained using a straight-forward maximum likelihood approach. While the original work defines a specific autoregressive order to model images effectively, we use a simplified version of RealNVP in this paper to model non-image data and pick the autoregressive order at random.

# 3 UNIVERSAL VALUE DENSITY ESTIMATION

## 3.1 ADDRESSING HINDSIGHT BIAS

Before we introduce our approach to estimating GCVFs corresponding to fixed or goal-conditioned policies, written $\tilde{Q}_\omega$ or $Q_\omega$ respectively, we will first point out the issues that arise when using common temporal-difference learning approaches. To efficiently approximate a GCVF, we have to address the challenge of learning from sparse rewards. Each TD update requires us to sample a goal and if this is done independently from the trajectory is unlikely to be a goal that has been reached. Andrychowicz et al. (2017) address this with a heuristic that oversamples achieved goals. This allows the agent to learn from failures and provides a way to speed up the training of a UVFA; however, it is a heuristic approach and can lead the agent to converge to non-optimal policies in stochastic domains. We illustrate this with a gridworld-example in Figure 1: here, the agent has to walk around a cliff to reach the goal in the bottom

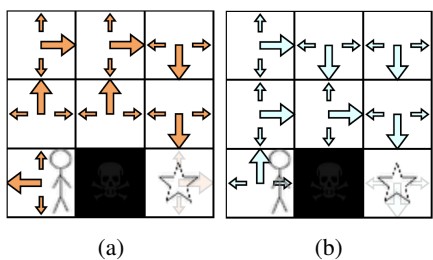

Figure 1: Simple cliff-walk domain. The agent starts in the bottom left, has to avoid the cliff depicted in black and reach the state in the bottom right. **a)** depicts the transition probabilities of an optimal policy while **b)** depicts a sub-optimal policy learned using HER.

right but may fall down the cliff due to environment noise affecting its movements. The optimal policy is to take a safe path to avoid falling down the cliff by accident; however, using HER we only learn about the true goal based on successful roll-outs and disregard episodes where the environment noise led to failure, i.e. the agent never learns from transitions leading down the cliff, underestimates their probability and therefore incorrectly chooses the shorter path.

Mathematically, this hindsight bias manifests itself as being equivalent to changing the environment dynamics. We examine the effect of changing the goal-sampling distribution in a hindsight temporal difference update rule. Using $\overline{\omega}$ to refer to the parameters of the previous iteration, the temporal difference update rule takes the form $\omega \leftarrow \overline{\omega} + \alpha\Delta$, with

$$\Delta := \int \rho^{\mu_\theta^g}(s,a)p(s'|s,a)p(g)\nabla_\omega Q_\omega(s,a;g)\delta ds,a,s',g$$

$$\delta := r^g(s,a) + \gamma Q_{\overline{\omega}}(s',\mu_\theta^g(s')) - Q_{\overline{\omega}}(s,a). \tag{1}$$

Altering $p(g)$ independently would be akin to using out-of-distribution samples in regression, can be addressed with additional representational capacity and the effect is little different from using using a replay buffer (Munos et al., 2016; Fujimoto et al., 2018); however, HER alters the distribution of goals $p(g)$ to be dependent on the observed transitions, i.e. transitions are sampled from $p(s'|s,a)p(g|s,a,s')$. This factorization can alternatively be written as $p(g|s,a)p(s'|s,a,g)$ which makes it plain how HER is modifying the transition dynamics and therefore the MDP that the agent is trying to solve. This is the source of hindsight bias in stochastic domains as observed in Figure 1.

## 3.2 VALUE DENSITY ESTIMATION

Knowing hindsight samples to be a good solution to recover a dense learning signal but to lead to an inconsistent estimator when combined with temporal-difference learning, we now show that density estimation can be used to estimate GCVFs using a sampling procedure similar to that of HER. We consider the special case where the task is for the agent to reach a given goal state. This scenario is common in goal-conditioned RL and also the one being addressed by Andrychowicz et al. (2017). In discrete environments, we can define such tasks by a reward that is positive if the goal is achieved and 0 otherwise. We define: $r^g(s,a) := (1-\gamma)\mathbb{1}(h(s,a) = g)$, where $h$ is a function that defines the achieved goal for any given state-action pair. In slight abuse of notation[2], we extend this definition to continuous environments and write $r^g(s,a) = (1-\gamma)\delta_{h(s,a),g}$. We can now show that the Q-function of such tasks forms a valid density function. Specifically, we notice that the Q function is equivalent

---

[2]Formally, the reward would have to be defined to be non-zero only in an $\epsilon$-ball around $h(s,a)$ and inversely proportional to the volume of this ball. All results hold in the limit $\epsilon \to 0$.

to the discounted probability density over goals that the agent is likely to achieve when following its policy, starting from the given state-action pair:

$$
\begin{aligned}
Q_{r^g}^{\mu}(s, a) &= \mathbb{E}\left[\sum_{t=0}^{\infty} \gamma^t r(s_t, a_t) | s_0 = s, a_0 = a, \mu\right] \\
&= (1 - \gamma) \sum \gamma^t \int p^{\mu}(s, a \xrightarrow{t} s') \delta_{h(s', \mu(s')), g} ds' =: F_{\gamma}^{\mu}(g|s, a)
\end{aligned}
\tag{2}
$$

It follows that we can learn an approximation of the goal-conditioned Q-function $Q_{r^g}^{\mu}(s, a)$ by approximating the value density $F_{\gamma}^{\mu}(g|s, a)$. This can be done using modern density estimators such as RealNVPs (see Section 2.4). To train the model, we gather training samples from a roll-out $s_0, a_0, s_1, a_1, \ldots$, collecting state-action pairs $s = s_t, a = a_t$ at random time-steps $t$ as well as future achieved goals $g = h(s_{t+j}, a_{t+j}); j \sim Geom(1 - \gamma)$. Notice that the above derivation assumes a fixed, i.e. not goal-conditioned policy. We relax this assumption in Section 3.4.

### 3.3 COMBINING ESTIMATORS

Learning a model of $F_{\gamma}^{\mu_{\theta}}$ already provides a valid estimator of the Q-function; however, relying on density estimation alone is insufficient in practice. As the discount factor approaches 1, estimating Q purely based on density estimation would require state-action-goal triplets that are hundreds of time-steps apart, leading to updates which suffer from extremely high variance. In practice, we therefore wish to limit the time-horizon of the density estimator while relying on temporal difference (TD) learning to backup values over long time horizons with low variance. Note that temporal-difference learning alone is insufficient in this scenario as it may never observe a reward[3]. Here, we propose to approximate the value by combining the value density estimator with temporal difference learning, using the density estimate as a lower bound on the target value. If the goal is within the time-horizon of the density estimator, the (scaled) density estimate approaches the true value and the bounded temporal difference loss simply matches Q to this density estimate. If the goal is outside its time horizon, the lower bound approaches 0 and we are using plain temporal difference learning to back up the value. The lower-bounded temporal difference loss is given by

$$
L(\omega) := \left(r^g(s, a) + \gamma \overline{Q} - \tilde{Q}_{\omega}(s, a; g)\right)^2, \qquad \overline{Q} := \max\left(\tilde{Q}_{\overline{\omega}}(s, a; g), F_{\Phi}(g|s, a)\right), \quad (3)
$$

where $F_{\Phi}(g|s, a)$ is a learned estimate of the density defined in Eq. 2.

### 3.4 LEARNING GOAL-CONDITIONED POLICIES

In Section 2.2, we introduced notation to differentiate between learning a model $\tilde{Q}$, which uses a goal-conditioned reward and a fixed policy and a model $Q$, where both quantities are conditioned on the goal. While the previous section gives us the means to efficiently train $\tilde{Q}$, goal-conditioned RL requires us to learn $Q$. Using a goal-conditioned policy in Eq. 2, we can write down a corresponding equivalence to a predictive long-term generative model: $F_{\gamma}^{\mu^{\overline{g}}}(g|s, a) := Q_{r^g}^{\mu^{\overline{g}}}(s, a)$. Here, $F_{\gamma}^{\mu^{\overline{g}}}(g|s, a)$ is the distribution of goals $g$ that the agent will reach when it starts in state $s$, takes action $a$ and then tries to reach the goal $\overline{g}$. As before, we can model this with a normalizing flow; however, in this case the model has to be conditioned on the intended goal $\overline{g}$ and takes the form $F_{\Phi}(g|s, a, \overline{g})$. Using this density estimate, we can alter the temporal difference target in Eq. 3 to efficiently train a UVFA:

$$
\overline{Q} := \max\left(Q_{\overline{\omega}}(s, a; g), F_{\Phi}(g|s, a, g)\right) \tag{4}
$$

This yields an iterative algorithm for goal-conditioned RL (Algorithm 1) that is efficient and avoids hindsight bias. Implementation details are discussed in Appendix B.

## 4 VALUE DENSITY IMITATION LEARNING

We now turn our attention to the problem of sample-efficient imitation learning. We wish to train the agent to imitate an expert's policy using only a few demonstration samples from the expert.

---

[3]If the environment is continuous and stochastic and the reward is as defined above, the agent always observes a reward of 0 on random samples.

**Algorithm 1** Universal Value Density Estimation

**function** UVD
 **for** $i \leftarrow 0..\#\text{Iterations}$ **do**
  Fill replay buffer with experience
  **for** $s, a, \overline{g}$ sampled from short replay buffer **do**
   Sample $t \sim \text{Geom}(1 - \gamma)$
   Sample achieved $g$ $t$ steps ahead of $s$
   Update $F_\Phi$ with $-\nabla_\Phi \log F_\Phi (g|s, a, \overline{g})$
  **for** $s, a, s', \overline{g}$ sampled from long replay buffer **do**
   $\overline{F} \leftarrow F_\Phi \left( \overline{g}|s', \mu_\theta^{\overline{g}}(s'), \overline{g} \right)$
   $\overline{Q} \leftarrow \max \left( \overline{F}, Q_{\overline{\omega}} \left( s', \mu_\theta^{\overline{g}}(s') \right) \right)$
   Update $Q_\omega$ with $\nabla_\omega \left( r^{\overline{g}}(s, a) + \gamma \overline{Q} - Q_\omega (s, a; \overline{g}) \right)^2$
   Update $\mu_\theta^{\overline{g}}$ with $\nabla_a Q_\omega (s, a; \overline{g}) \big|_{a = \mu_\theta^{\overline{g}}(s)} \nabla_\theta \mu_\theta^{\overline{g}}(s)$

**Algorithm 2** Value Density Imitation

**function** VDI
 **for** $i \leftarrow 0..\#\text{Iterations}$ **do**
  Fill replay buffer with experience
  **for** $s, a$ sampled from short replay buffer **do**
   Sample $t \sim \text{Geom}(1 - \gamma)$
   Sample target states $\overline{s}$ $t$ steps ahead of $s$
   Update $F_\Phi$ with $-\nabla_\Phi \log F_\Phi (\overline{s}|s, a)$
   Update $d_\Phi$ with $-\nabla_\Phi \log d_\Phi (s)$
  **for** $s, a, s'$ sampled from long replay buffer **do**
   Sample $\overline{s}$ uniformly from expert data
   $\overline{Q} \leftarrow \max \left( F_\omega (\overline{s}|s, a), \gamma \tilde{Q}_{\overline{\omega}} (s', \mu_\theta (s')) \right)$
   Update $\tilde{Q}_\omega$ with $\nabla_\omega \left( \overline{Q} - \tilde{Q}_\omega (s, a; \overline{s}) \right)^2$
  **for** $s, a$ from long replay buffer **do**
   Sample $\overline{s}$ from expert data with $p = \frac{1}{d_\Phi(\overline{s})}$
   Update $\mu_\theta$ with $\nabla_a \tilde{Q}_\omega (s, a; \overline{s}) \big|_{a = \mu_\theta(s)} \nabla_\theta \mu_\theta (s)$

Different formulations exist to solve this problem (see Section 2.3), but the formulation which has arguably been the most promising is to train the agent to explicitly match the expert's state-action distribution or occupancy measure (Ziebart et al., 2008; Ho and Ermon, 2016). Our next step is therefore to extend our findings from the previous section to state-action or state distribution matching. In the literature, this is commonly framed as a divergence minimization problem with recent work investigating different measures (Ghasemipour et al., 2020; Ke et al., 2019); however, this view is incomplete as the divergence measure has to be approximated and the quality of the approximation can be significantly more impactful than the choice of measure. Similar to Schroecker and Isbell (2017); Schroecker et al. (2019), we propose to estimate the gradient $\nabla_\theta \log d^{\mu_\theta}(\overline{s})$ from self-supervised data (where we write $\overline{s}$ to refer to demonstrated states). With an estimate of this gradient, we can maximize the likelihood of expert state-action pairs which gives a straight-forward approach to minimizing the KL-divergence between the two occupancy distributions. Note that while the gradient is evaluated at expert states, the derivative itself depends only on the agent's behavior. Unlike the discriminator in an adversarial approach, the quality of the approximation therefore does not depend on the number available expert samples. Using the state-distribution gradient alone in a maximum-likelihood algorithm disregards expert actions, but is optimal if the expert's behavior can be uniquely described using a reward-function that depends only on the current state. We will evaluate state-distribution matching as imitation from observation in Section 5.2. To extend the approach to unambiguous state-action distribution matching, we can combine it with the behavioral cloning gradient as $\nabla_\theta \log \rho^{\pi_\theta}(\overline{s}, \overline{a}) = \nabla_\theta \log \pi_\theta(\overline{a}|\overline{s}) + \nabla_\theta \log d^{\pi_\theta}(\overline{s})$, where we assume a stochastic policy $\pi_\theta$ in place of a deterministic one. We found, however, that the behavioral cloning gradient can dominate a noisy estimate of the state-distribution gradient and lead to overfitting. Instead, we propose to augment the state to include the previous action that lead to the state. This approach attempts to match the joint distribution of action and next state which implies the state-action-distribution based on environment dynamics.

If applied to a single state, following the state-distribution gradient teaches the agent to go to that state. It is thus no surprise that it can be shown to be equivalent to the policy gradient for the right goal-conditioned reward function (also see Schroecker et al., 2019). Specifically, the state-distribution gradient is equivalent to the weighted policy gradient in the average-reward setting (using $r^{\overline{s}}(s, a) = \delta_{s, \overline{s}}$). We have:

$$\nabla_\theta \log d^{\mu_\theta}(\overline{s}) = \frac{\nabla_\theta d^{\mu_\theta}(\overline{s})}{d^{\mu_\theta}(\overline{s})} = \nabla_\theta \frac{\int d^{\mu_\theta}(s) \delta_{s, \overline{s}} ds}{d^{\mu_\theta}(\overline{s})} = \frac{\nabla_\theta J^{r^{\overline{s}}}(\theta)}{d^{\mu_\theta}(\overline{s})} \tag{5}$$

Intuitively, the policy gradient leads the agent toward a demonstration state $\overline{s}$, while the weight ensures that all demonstration states are visited with equal probability. This gives rise to Value Density Imitation Learning: Using self-supervised roll-outs, the algorithm learns a goal-conditioned Q-function as well as an unconditional density estimator $d_\omega(s)$ of the agent's state-distribution. Next,

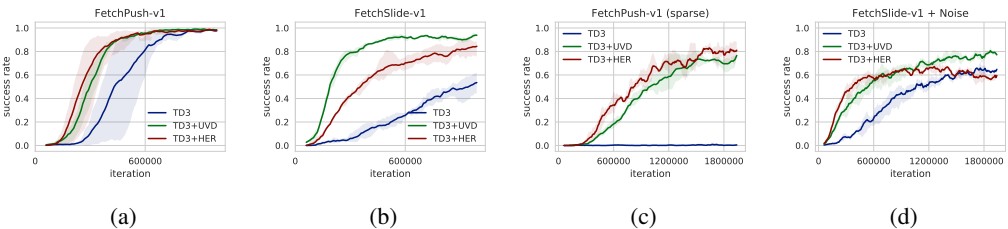

(a)                (b)                (c)                (d)

Figure 2: Average and range of success rate across 3 seeds of TD3, UVD and HER on variations of the Fetch manipulation domains. *FetchPush* and *FetchSlide* correspond to the original domains. *Fetch-Push (Sparse)* shows the advantage of hindsight updates when the required precision is significantly higher while *FetchSlide + Noise* shows the effect of hindsight bias in stochastic domains.

demonstration states are sampled with probability proportional to $\frac{1}{d_\omega(\bar{s})}$ and used as goals when estimating the policy-gradient (see Alg. 2 for details).

## 5 EXPERIMENTS

### 5.1 GOAL-CONDITIONED REINFORCEMENT LEARNING

We first evaluate UVD on a suite of simulated manipulation tasks involving a Fetch robot arm. This suite has been developed by Andrychowicz et al. (2017) to show the strengths of hindsight experience replay. Our goal here is twofold: 1. We show that UVD matches the performance of HER on the unmodified deterministic tasks where HER does not suffer from hindsight bias. 2. We show that HER converges to a sub-optimal policy on a stochastic variation of the task while UVD is still able to solve the task efficiently. We compare all methods using the same hyper-parameters (see Appendix A).

The first domain in the suite, *FetchPush*, requires the robot arm to learn to push an object to any given target location. The reward signal is sparse as the agent is given a non-zero reward only if the object reaches the desired location. In Figure 2a, we can see that TD3 with HER and TD3 with UVD perform similarly. Contrary to the original findings by Andrychowicz et al. (2017), we also find unmodified TD3 to be able to solve the task accurately using roughly twice the amount of training samples. This indicates that the area around the goal in which the object is considered to be at the desired location is relatively large. If we apply a stricter criterion for the goal being reached by reducing the size of the goal area by a factor of 100, we can see that hindsight samples are necessary to learn from sparse rewards. In this variant, TD3 fails to learn a useful policy (see Figure 2c).

The second domain in the suite, *FetchSlide*, requires the robot to slide the object toward a desired location that is out of reach of the robot arm. In Figure 2b, we can see UVD and HER learning to solve the task quickly while TD3 without hindsight samples requires significantly more training samples. Unlike in the case of *FetchPush*, TD3+UVD learns slightly faster in this domain than TD3+HER but both are able to solve the task eventually. Both, FetchSlide and FetchPush, are deterministic domains that play to the strengths of hindsight experience replay. In practice, however, manipulation with a real robot arm is always noisy. In some cases, HER can overcome this noise despite suffering from hindsight bias; however, this is not always the case. Here, we introduce a variation of the FetchSlide domain which adds Gaussian noise to the actions of the agent. We scale this noise based on the squared norm of the chosen actions $\frac{1}{2e}||\max(0, a - 0.5 \cdot \mathbf{1})||_2^2$. This scaling allows the agent to adapt to the noise; however, doing so requires the agent to accurately assess the risks of its actions. In Figure 2d, we can see that while HER initially learns quickly as in the deterministic domain, it converges to a policy that is noticeably worse than the policy found by TD3+UVD. This shows the effect of hindsight bias: in the presence of noise, the agent assumes the noise to be benign. UVD, on the other hand, estimates the risk accurately and achieves a higher success rate than HER.

### 5.2 IMITATION LEARNING

We compare the demonstration-efficiency of Value Density Imitation with GAIL, on common benchmark locomotion tasks (Brockman et al., 2016). We compare against GAIL as no method has

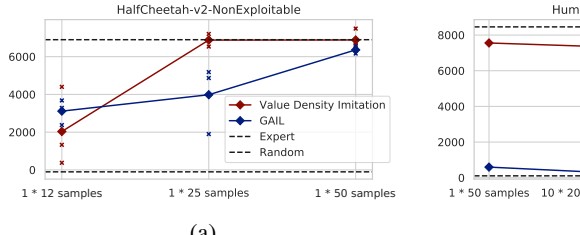

Figure 3: Comparison of GAIL and VDI on variations of standard benchmark tasks. The x-axis shows the number of demonstrated trajectories times state-action pairs per trajectory. VDI requires fewer demonstrations compared to GAIL.

reported better demonstration-efficiency on these domains to date. Recently, the suite of locomotion tasks has become the go-to benchmark for imitation learning (e.g. Ho and Ermon, 2016); however, we find that the unmodified locomotion tasks, despite their popularity, are easily exploited in an imitation learning context and are a bad measure of whether an algorithm imitates the expert or not. While it is impressive that GAIL is able to solve *Humanoid-v2* with less than a dozen of disjointed demonstrated states, it is also clear that the agent is not learning to imitate the motion itself. The dominating source of reward in these tasks comes from the velocity in a particular direction. This is problematic as the velocity is fully observable as part of the state and, in the case of humanoid, may be encoded in more than one of the features found in the state-vector. Even the simple average of the state-features may be equivalent to a noisy version of the original reward signal. Moreover, as the reward is a linear combination of state-features, we know that accurate distribution matching is not necessary and matching feature expectations is sufficient (Abbeel and Ng, 2004; Ho and Ermon, 2016). To alleviate this, we remove task-space velocities in $x, y$ directions from the state-space. A second source of bias can be found in the termination condition of the locomotion domains. Kostrikov et al. (2019) point out that GAIL is biased toward longer trajectories and thus tries to avoid termination, which in the case of locomotion means to avoid falling. While Kostrikov et al. adjust the algorithm itself to avoid such bias, we instead propose to remove the termination condition and use an evaluation which cannot be exploited by a biased method. We find that other methods such as DAC (Kostrikov et al., 2019) may require re-tuning to solve these tasks but find that GAIL still performs well on the modified tasks. This is unsurprising as GAIL has been applied to real-world applications (e.g. Wang et al., 2017).

We focus on two locomotion tasks in particular: *HalfCheetah-v2* and *Humanoid-v2*. *HalfCheetah-v2* is comparatively easy to solve while providing a high skill ceiling. With the original threshold for solving the task being set at a score of 4500, recent advances in reinforcement learning train policies that achieve 3-4 times as high a score (Fujimoto et al., 2018; Haarnoja et al., 2018) (removing velocity from the state reduces the top-speed achieved by the TD3-trained expert). Our second domain of choice is Humanoid-v2, which is generally considered to be the most complex locomotion task. Unlike in the *HalfCheetah-v2* domain, learning to move without falling can be a significant challenge for a learning agent. We furthermore find it sufficient to match state-distributions to solve *HalfCheetah-v2* and thus teach the agent from observation only when using VDI. In the case *Humanoid-v2*, we find that demonstrated actions significantly aid exploration and thus include them. The results can be seen in Figures 3a and 3b as well as in Appendix C. While both methods are able to achieve near-expert performance on *HalfCheetah-v2-NonExploitable* using a single demonstrated trajectory sub-sampled at the same rate as used by Ho and Ermon, we find VDI to be able to imitate the expert if the trajectory is sub-sampled even further. On *Humanoid-v2-NonExploitable*, we find the difference to be more drastic: while GAIL is able to learn locomotion behavior from a similar number of trajectories as used in the original paper (but using more state-action pairs), the performance drops off quickly if we reduce the number of trajectories further. VDI is able to learn from only a single demonstration. Both methods are able to achieve great demonstration-efficiency by matching the expert's state-action distribution; however, GAIL does so by learning a distance function using demonstrations as training samples. As the number of demonstrations shrinks, learning a good discriminator to serve as a reward becomes more difficult. VDI side-steps this issue by learning a goal-conditioned Q-function based on self-supervised roll-outs alone. The demonstrations are used to condition the Q-function and don't serve directly as training data for any network.

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

# A HYPERPARAMETERS

In Table 1, we list the hyper-parameters used for TD3, TD3+HER and TD3+UVD on the Fetch experiments. When applicable, we largely use the same hyper-parameters for each algorithm. Two important exceptions are the learning rate which is tuned individually for each algorithm (TD3+UVD benefits reliably from higher learning rates whereas TD3 and TD3+HER does not always converge to a good solution at higher learning rates) as well as the output activation of the Q-network. A tanh activation is used to scale the value to the range of -50 to 50 in the case of TD3 and TD3+HER as we found this to drastically improve performance. Since the density is not bounded from above, the same activation cannot be used in the case of TD3+UVD.

In Table 2, we list the hyper-parameters used for VDI in the locomotion experiments. Parameters are largely identical between environments; however, in some cases we trade off higher learning speed for reduced accuracy on *HalfCheetah*. In the case of GAIL, we use the implementation found in OpenAI baselines[4], using 16 parallel processes (collecting 16 trajectories per iteration) and do not modify the parameters.

Table 1: Common parameters in Fetch environments

| **General parameters** | |
| --- | --- |
| Environment steps per iteration | 1 |
| $\gamma$ | 0.98 |
| Batch size | 512 |
| Replay memory size | 1500000 |
| Short replay memory size | 50000 |
| Sequence Truncation (Density estimation training) | 4 |
| Optimizer | Adam |
| **Policy** | |
| Hidden layers | 400, 400 |
| Hidden activation | leaky relu |
| Output activation | tanh |
| Exploration noise $\sigma$ | 0.1 |
| Target action noise $\sigma$ | 0.0 |
| Learning rate | $2 \cdot 10^{-4}, 8 \cdot 10^{-4}$ (TD3+UVD) |
| **Q-network** | |
| Hidden layers | 400, 400 |
| Hidden activation | leaky relu |
| Output activation | 50 tanh, linear (TD3+UVD) |
| Learning rate | $2 \cdot 10^{-4}, 8 \cdot 10^{-4}$ (TD3+UVD) |
| **RealNVP** | |
| Bijector hidden layers | 300, 300 |
| Hidden activation | leaky relu |
| Output activation, scale | tanh(log()) |
| Output activation, translate | linear |
| num bijectors | 5 (slide), 6 (push) |
| Learning rate | $2 \cdot 10^{-4}$ |

---

[4]https://github.com/openai/baselines/tree/master/baselines/gail

Table 2: VDI parameters in locomotion environments

| Parameter | HalfCheetah | Humanoid |
|---|---|---|
| **General parameters** | | |
| Environment steps per iteration | 10 | 10 |
| $\gamma$ | 0.995 | 0.995 |
| Batch size | 256 | 256 |
| Replay memory size | 1500000 | 1500000 |
| Short replay memory size | 500000 | 500000 |
| Sequence Truncation (Density estimation training) | 4 | 4 |
| Optimizer | Adam | Adam |
| **Policy** | | |
| Hidden layers | 400, 300 | 400, 300 |
| Hidden activation | leaky relu | leaky relu |
| Output activation | tanh | tanh |
| Exploration noise $\sigma$ | 0.3 (until iteration 25000), 0.1 (after) | 0.1 |
| Target action noise $\sigma$ | 0.0 | 0.0 |
| Learning rate | $3 \cdot 10^{-4}$ | $3 \cdot 10^{-4}$ |
| **Q-network** | | |
| Hidden layers | 400, 400 | 400, 400 |
| Hidden activation | leaky relu | leaky relu |
| Output activation | linear | linear |
| Learning rate | $3 \cdot 10^{-4}$ | $1 \cdot 10-4$ |
| **RealNVP** | | |
| Bijector hidden layers | 400, 400 | 400, 400 |
| Hidden activation | leaky relu | leaky relu |
| Output activation, scale | tanh(log()) | tanh(log()) |
| Output activation, translate | linear | linear |
| num bijectors | 5 | 5 |
| Learning rate | $1 \cdot 10^{-4}$ | $2 \cdot 10^{-5}$ |
| L2-regularization | $1 \cdot 10^{-5}$ | $1 \cdot 10^{-6}$ |
| Spatial smoothing | 0.1 | 0.1 |
| Temporal smoothing | 0. | 0.98 |

## B    PRACTICAL CONSIDERATIONS

There are a number of implementations decisions that were made to improve the sample-efficiency and stability of Value Density Estimation and its application to imitation learning. Here, we review these decisions in more detail.

### B.1    UNIVERSAL VALUE DENSITY ESTIMATION

**Using an exploration policy:**    In most cases self-supervised roll-outs will require the agent to explore. In our method, we combine a temporal difference update rule as is usually found in deterministic policy gradients with density estimation. While the temporal-difference update rule can handle off-policy data from an exploration policy, density-estimation is on-policy. In practice, however, Fujimoto et al. (2018) add Gaussian noise to the target-Q function and report better results by learning a smoothed Q-function that is akin to the on-policy Q-function with Gaussian exploration noise. It thus stands to reason that we can omit off-policy correction in the density-estimator.

**Truncating the time horizon:**    The training data for learning a long-term model can be fairly noisy. While we can expect density estimation to be efficient over a horizon of just a few time-steps, the variance increases dramatically as $\gamma$ increases. This is the primary motivation for utilizing temporal-difference learning in conjunction with universal value density estimation. To better facilitate stable training, we truncate the time-horizon of the density-estimator to a fixed number of time-steps $T$. The temporal-difference learning component is thus solely responsible for propagating the value beyond this fixed horizon. The effect this has on the optimal policy is small: the Q-value will be underestimated by ignoring visitations with time-to-recurrence greater than $T$. A greater time-horizon

boosts the effect of hindsight samples and leads to a learning signal that is less sparse while a smaller time-horizon reduces the variance of the estimator.

**Using a replay buffer:**  Using a replay buffer is essential for sample-efficient training with model-free reinforcement- and imitation-learning methods and improves the stability of deterministic policy gradients. We find that this is true for training long-term generative models as well. While importance sampling based off-policy correction for density estimation is possible, we find that it introduces instabilities and is thus undesirable. Instead, we propose to use a separate, shorter replay-buffer for density estimation to mitigate the undesirable effects of off-policy learning while retaining some of the benefits.

**Delayed density updates:**  Temporal-difference learning with non-linear function approximation is notoriously unstable. To help stabilize it, a common (Mnih et al., 2015; Fujimoto et al., 2018) trick is delay the update of the target network and allow the Q-function to perform multiple steps of regression toward a fixed target. Since we use the long-term predictive model $F_\omega$ to calculate the temporal-difference regression target, we apply the same trick here. We maintain a target network $F_{\overline{\omega}}$ which we set to be equal to the online density estimator $F_\omega$ after a fixed number of iterations. In Value Density Imitation, we use the same procedure to maintain a frozen target network of the unconditional state density estimator $d_\omega$.

**Normalizing states:**  As our method depends on density estimation, the resulting values are heavily affected by the scale of the features. We therefore normalize our data based on the range observed in random roll-outs as well as, in the case of imitation learning, the range seen in the given demonstrations.

### B.2    VALUE DENSITY IMITATION

**Averaging logits:**  While the dimensionality of the goal in goal-conditioned reinforcement learning is typically small, Value Density Imitation requires us to use the entire state as a goal. This, however, can be difficult if the number of features is large. If the state-features are independent, the density suffers from the curse of dimensionality as it is multiplicative and the Q-values will be either extremely large or extremely small. Even if the true density function does not exhibit this property, Normalizing Flows predict the density as a product of $N$ predicted logits and prediction errors are therefore multiplicative. To combat this, we take the average of the predicted logits rather than the sum, effectively taking the $N-th$ square root of the Q-function. We find that this approximation works well in practice and justify it further in appendix D.

**Bounding weights on demonstrated states:**  In Value Density Imitation, we sub-sample demonstrations states proportional to $\frac{1}{d_\omega(\overline{s})}$ to ensure demonstration states to be visited with equal probability. In this formulation, demonstration states that are especially difficult to reach may be over-sampled by a large factor and destabilize the learning process. To counteract this, we put a bound on the weight of each demonstration state: for each batch, the weights are normalized and an upper bound is applied.

**Spatial and temporal smoothing:**  We apply two kinds of smoothing to the learned UVFA to improve the stability of the learning algorithm. Spatial smoothing ensures that similar state-action pairs have similar value and is achieved by adding Gaussian noise to training samples of the target state when training the density estimator. Temporal smoothing ensures that the learned value does not spike too suddenly. Instead of using $F_\gamma(\overline{s})$ as a temporal difference regression target, we use a mixture of the density estimation and the temporal-difference lower bound. Using a temporal smoothing factor $\lambda$, the full temporal difference loss is then given by:

$$L(\omega) := \left( r^g(s,a) + \gamma\overline{Q} - \tilde{Q}_\omega(s,a;g) \right)^2,$$
$$\overline{Q} := \lambda\tilde{Q}_{\overline{\omega}}(s,a;g) +$$
$$(1-\lambda)\max\left( \tilde{Q}_{\overline{\omega}}(s,a;g), F_\Phi(g|s,a) \right)$$

## C   ADDITIONAL EXPERIMENT DETAILS

Figure 4: Individual learning curves for each imitation learning experiment.

In Section 5.2, we describe imitation learning experiments on non-exploitable versions of the commonly used *HalfCheetah* and *Humanoid* benchmark tasks. Figure 4 shows the individual learning curves for both GAIL and VDI on each domain. Note, that the number of environment steps for GAIL is an order of magnitude larger than on VDI and several times higher than what is necessary to solve the unmodified, exploitable variant of these domains. Because it takes several days to collect this amount of experience for each run, we terminate some runs early if it is clear they have converged and if at least 75 or 150 million environment steps have been taken in each respective domain. While more sample-efficient versions of GAIL have been proposed (Kostrikov et al., 2019) and work well on the original formulation of the domains, we were unable to achieve good results on the corrected version of the domains. It is likely that the large amount of regularization imposed on the discriminator in Discriminator-Actor-Critic makes this formulation more likely to exploit the velocity given as part of the observation space and that significant effort would have to be put in to find the right hyper-parameters to apply this method on the non-exploitable domains.

## D   ESCAPING THE CURSE OF DIMENSIONALITY

In section 3, we introduced a method which uses the probability density predicted by a normalizing flow as a Q function. We showed that this density Q function is a valid Q function with a corresponding reward function that is sensible for many practical task. In this appendix, we consider the numerical properties of the universal value density estimator and propose a slight variation that is easier to handle numerically. To see the numerical challenge in using density estimators as value functions, we take a look another look at a single bijector of a RealNVP; here, the bijector $f_\omega(z) = (s_\omega(z), t_\omega(z))$ is predicting an affine transformation of $z \sim p_z(\cdot) = \mathcal{N}(0, I)$ to $x \sim p_x(\cdot)$, i.e. $x_i = s_\omega(z)_i z_i + t_\omega(z)_i$.

The log-density of x is then given as $p_x(x) = e^{\sum_{i=0}^{N} s_\omega^{-1}(x) - \left( \frac{x_i - t_\omega^{-1}(x)_i}{s_\omega^{-1}(x)_i} \right)^2 + \log \frac{1}{\sqrt{2\pi}}}$. It is readily apparent that this value can easily explode, especially when used as a target Q value in the mean-squared loss of a temporal difference update. To combat this, we propose to scale the logits with the

dimensionality $N$, i.e. we use $p_x(x) = e^{\frac{1}{N} \sum_{i=0}^{N} s_\omega^{-1}(x) - \left( \frac{x_i - t_{\bar{\omega}}^{-1}(x)_i}{s_{\bar{\omega}}^{-1}(x)_i} \right)^2 + \log \frac{1}{\sqrt{2\pi}}}$. As this corresponds to only a constant factor on $\log F_\gamma$, the gradient-based density estimation is not affected.

While the change is simple, average logits instead of taking the sum, we need to justify the approximation anywhere the value density and the Q-function are used: first, we show that $J(\theta)^{\frac{1}{N}}$ can be used in place of $J(\theta)$ in both, goal-conditioned reinforcement learning and in imitation learning without changing the optimal policy; second, we justify using $Q_{r^g}^{\mu_\theta}(s, a)^{\frac{1}{N}}$ in place of $Q_{r^g}^{\mu_\theta}(s, a)$ when computing the policy gradient; and, finally, we show that we can justify the use of the N-th root in the temporal difference learning update.

**Using the scaled objective $J(\theta)^{\frac{1}{N}}$:** Here, we have to consider two cases. In the case of goal-conditioned RL, we have to show that $\max_\theta J(\theta) = \max_\theta J(\theta)^{\frac{1}{N}}$. To this end, it is sufficient to note that the reward function is strictly non-negative and thus $(\cdot)^{\frac{1}{N}}$ is a monotonous function. In the case of imitation learning, we can immediately see that the change corresponds to a constant factor on the state-distribution gradient:

$$\frac{1}{N} \nabla_\theta \log d^{\mu_\theta}(\bar{s}) = \frac{\nabla_\theta J_{r^{\bar{s}}}(\theta)^{\frac{1}{N}}}{d^{\mu_\theta}(\bar{s})^{\frac{1}{N}}} \tag{6}$$

**Estimating the policy gradient $\nabla_\theta J(\theta)^{\frac{1}{N}}$:** The deterministic policy gradient theorem (Silver et al., 2014) shows that maximizing the Q-value in states sampled from the agent's discounted on-policy state-distribution is equivalent to maximizing the reinforcement-learning objective. This is immediately apparent if the representation of the policy is sufficiently expressive and agent is able to take the action with maximum value in every state. Due to monotonicity of $(\cdot)^{\frac{1}{N}}$, this is true when using $Q_\Phi(s, a; g)^{\frac{1}{N}}$ as well. If the policy is not able to maximize the Q-function everywhere, the deterministic policy gradient theorem shows that sampling from the discounted state-distribution leads to the agent making the right trade-offs. Using $Q(s, a; g)^{\frac{1}{N}}$, however, leads to a different trade-off. In practice, this is typically ignored: deterministic policy gradients used off-policy with a replay-buffer are not guaranteed to make the right trade-off and even if they are used on-policy, the discount-factor is typically ignored and the resulting estimate of the policy gradient is biased (Nota and Thomas, 2019).

$Q_g^\mu(s, a)^{\frac{1}{N}}$ **as TD target:** Finally, we need to show that we can use the N-th root in the computation of the temporal difference learning target. To this end, we make use of the fact that the reward in continuous environments can be assumed to be 0 when computing the temporal difference target. In this case $(r^g(s, a) + \gamma Q_{r^g}^\mu(s', \mu(s')))^{\frac{1}{N}}$ becomes $\gamma^{\frac{1}{N}} Q_{r^g}^\mu(s', \mu(s'))^{\frac{1}{N}}$ and increasing $\gamma$ is sufficient to compute a valid regression target for $Q_{r^g}^\mu(s, a)^{\frac{1}{N}}$

