# OpenReview forum: "Universal Value Density Estimation for Imitation Learning and Goal-Conditioned Reinforcement Learning"
_ICLR.cc/2021/Conference — Reject_

### Official Review · AnonReviewer4 · 2020-10-27
**Interesting work, but limited experimental results and unclear contribution**

**Rating:** 5
**Confidence:** 4

**Review:**

This paper proposes a method for solving (state-distribution matching) imitation learning and goal-conditioned reinforcement learning (GCRL) by training a value function both with standard TD learning and Monte Carlo (MC) learning. They point out that for the case of indicator rewards, the Q function corresponds to a density model over the discounted state occupancy measure, and so they propose to train the MC-Q function by training a normalizing flow via MLE. The authors propose to combine the estimates of the two during TD backups by taking the max of the two estimates, and demonstrate that the overall algorithm results in better performance on the Fetch GCRL tasks (relative to HER) and on “non-exploitable” versions of locomotion tasks which remove the velocity and termination information that past work on imitation learning has included.

Overall, the paper studies an interesting connection between GCRL and imitation learning, and the idea of training a goal-conditioned value function via MLE seems novel and intriguing. I have concerns that the contribution of the paper is a bit unclear, and the overall performance gains are rather small.

In more detail:

It’s unclear if the benefits are from the “Q function is a density model” perspective, or if the benefits are just coming from mixing TD and Monte-Carlo estimates together. For example, how would the performance change if the normalizing model F was simply replaced with a Q function trained on Monte-Carlo estimates?

There were a number of design decisions and claims that I did not understand. For one, I don’t understand why VDI is immune to hindsight bias. The authors have experiments that shows that there is some small performance gain when stochasticity is introduced, but it seems that the distribution of goals that F is trained on will similarly be biased. Another question is why the authors propose to approximate the two values by taking their max. Is there any theoretical reason why this would be desired? Lastly, what does it mean for a goal to be “within the time-horizon of the density estimator”? Is there a mathematical definition of this condition, and why taking the max would address this problem?

The related works section is generally good, though I recommend the authors also discuss the relationship to [1] which also uses the gradient of the state distribution to train the policy.

Lastly, perhaps the main weakness is that the performance gains seem relatively small, which makes the impact of the work relatively small given that the main contribution seems to be empirical.

[1] Hazan, Elad, et al. "Provably efficient maximum entropy exploration." International Conference on Machine Learning. 2019.


--- Post Rebuttal ---

I had some confusion and concerns about the paper. Most of the confusions were addressed and made me view the paper slightly more favorably. However, my main concern wasn't addressed. In particular, it's unclear if the benefits are because of the "Value Density Estimation" benefits or if it's because of the mix of non-bootstrap updates. Comparing to a method that mixes in more Monte Carlo estimates (e.g. n-step bootstrap) would address this concern. Theoretically, the authors suggest that the reward should be 1/epsilon with epsilon -> 0. However, it seems like this was not done in the experiments, and it's unclear how epsilon should be decayed in practice. Given these concerns, I will maintain my score.

---

> ### Author Response · Authors · 2020-11-16
> **Rebuttal**
>
> Thank you for your review, we hope to address your concerns in the following.
>
> Regarding why UVD is immune to hindsight bias, the bias is not a property of the data, it is a property of the algorithm which is completely different. In Equation 2, we showed that the value-function is equal to the probability density of achieved goals conditioned on the state, action and intended goal. The way the density estimator is trained is completely standard and known to be unbiased: we do maximum-likelihood on samples of achieved goals, state, action and intended goal. Hindsight bias appears in HER because the achieved goal is used to choose which value function to update after the bootstrap target is already given; nothing like this happens in UVD.
>
> Regarding the comparison to monte-carlo estimates, the difference between UVD and HER compared to regular TD learning or monte-carlo estimates is that the former can update their value functions even if the agent never observes a reward. In the FetchPush-sparse experiments this is approximately the case as the TD-learning agent only observes a reward once every few hundred thousand frames. While we can run monte-carlo experiments for comparison, it is simply implausible that the agent would be able to learn in this case. Note that if we make the reward-function even sparser, to the extent where the robot arm cannot achieve the necessary precision to observe reward, methods such as UVD and HER would still learn to approach the goal and get closer while conventional reinforcement learning methods simply have no signal at all.
>
> Regarding the max and the time-horizon, to train the density estimator, we collect triples of state, action and a future state based on a geometric distribution. We do indeed limit how far apart state and future state can be as the variance of the training samples would otherwise be incredibly large and it becomes difficult to train the density estimator. This is mentioned in the main body of the paper but further discussed in Appendix B.1. Mathematically, the sums in Eq. 2 are truncated to a finite horizon. The choice of the max is directly related: by truncating the sequence length, we are setting the reward of far future states to 0 and underestimating the value-function. Thus we use it as a lower bound and allow temporal-difference learning to take care of estimating the value over long horizons with low variance (which it was designed for).
>
> Regarding performance gains, we believe that the performance gains in the imitation learning setting are generally large. Where GAIL fails to learn at all with 10 demonstrations on humanoid, VDI is able to learn from just 1. In the goal-conditioned setting the performance gains are indeed more modest. This is because HER works surprisingly well empirically despite its theoretical flaws. We still believe that it is valuable to develop methods that do not provably converge to incorrect solutions as HER does.

---

> > ### Comment · AnonReviewer4 · 2020-11-22
> > **Re: Rebuttal**
> >
> > > Regarding why UVD is immune to hindsight bias...
> >
> > Thank you for clarifying.
> >
> > > Regarding the comparison to monte-carlo estimates, the difference between UVD and HER compared to regular TD learning or monte-carlo estimates is that the former can update their value functions even if the agent never observes a reward....
> >
> > I still don't understand how the UVD training procedure is different from a Monte-Carlo estimate of a universal value function approximator.
> >
> > Regardless of whether you use TD or Monte-Carlo, my question was asking about what you're estimating, and if it's different from a universal value function approximator. I think the difference that the UVD is learning a value function of the form Q(state, action, intended goal, achieved goal) rather than Q(state, action, goal), where Q(state, action, intended goal, achieved goal) = F(achieved goal | state, action, intended goal). Is that right? My point was that the training procedure for F seems like a Monte-Carlo way of estimating Q(state, action, intended goal, achieved goal) since you use samples rather than bootstraps.
> >
> > This also makes me think that there's a bit of a type error: For continuous state space, F is a probability density model over the state space. In that case how is it possible that F (a probability *density* model) can be equal to the Q function, which is the expectation of an indicator function--i.e., a probability and not a probability density? Does this equivalence only hold for discrete state spaces?
> >
> > >  The choice of the max is directly related: by truncating the sequence length, we are setting the reward of far future states to 0 and underestimating the value-function. Thus we use it as a lower bound and allow temporal-difference learning to take care of estimating the value over long horizons with low variance (which it was designed for).
> >
> > Thank you for explaining this.

---

> > > ### Author Response · Authors · 2020-11-23
> > > **Re: Re: Rebuttal**
> > >
> > > The learned function does not differ from a universal value-function approximator, but the training procedure is different. We train F(achieved goal|s, a, intended goal) via density estimation and show the following equivalence to UVFAs: Q(s, a, g) = F(achieved goal=g| s, a, intended goal=g)
> > >
> > > There is no bootstrapping, so it suffers from the same high variance that monte-carlo estimates have. That's precisely why we propose a method to combine it with temporal difference learning. If we could bootstrap the density estimator directly, we wouldn't need to combine it with a TD-learning approach. Nevertheless, it is not the same as value regression which is what we typically mean when we talk about monte-carlo estimation of the value function. Relabelling the goals as in HER is not valid in value regression either, but we can use the achieved goals to estimate the density F.
> > >
> > > As for the equivalence in continuous spaces, the dirac-delta indicator function is a density function. To give a little bit more of an explanation: if you define the reward function to be 1/epsilon in a ball with volume epsilon centered around the goal and 0 otherwise, then the corresponding Q-function approaches the value density function as eps->0.

---

### Official Review · AnonReviewer2 · 2020-10-28
**Solid method, more extensive experiments will be helpful**

**Rating:** 5
**Confidence:** 3

**Review:**

Summary: This paper focuses on goal-conditioned policy learning in the environment with stochastic dynamics and tries to address the bias of HER. Given a policy and a sparse reward function according to the goal,  the universal value function is converted to the discounted probability density over goals. And the authors use the recent work RealNVP about density estimation to learn the universal value function. The proposed value density estimation is also extended to imitation learning. Experiments show that the proposed method outperforms baselines in both goal-conditioned policy learning and imitation learning.

Clarity:
The clarity in writing can be improved in some points. First, it will be good if more details about RealNVP are introduced. How is it applied to learn the probability density? Second, in section 3.3, "if the goal is within the time-horizon of the density estimator....". Does it mean another hyper-parameter, e.g. time horizon of the density estimator, is introduced here? Is the proposed method sensitive to the value of this hyper-parameter? Third, in figure 1, do the arrows represent the learned policy with HER and UVD. It will be more convincing if the learning curves on this toy sample is also presented (in the appendix).

Originality:

Significance:
The proposed method is technically sound and experiments demonstrate that it performs well in comparison to the baselines in some continuous control tasks. My concern lies in the complexity of implementing the proposed method compared with the baselines, and whether the method can be easily used in other domains with rich observation space and discrete action space.

Pros:
*The proposed method is well-motivated to solve a problem of HER for goal-conditioned policy.
*The proposed method works well in some continuous control tasks and the authors carefully discuss the advantage of the proposed method in different environment settings (Figure 2) and with the different datasets (Figure 3).

Cons:
*In table 1 and table 2, I notice the use of RealNVP introduces some hyper-parameters. Is it easy to tune these hyper-parameters?
*Is the proposed method applicable to domains with discrete action space?
*In Appendix D, there is a discussion about dimensionality. The data samples on Mujoco has low-dimensional state features.  If the observation space is more complicated and rich (e.g. RGB images), could the proposed method be easily used for imitation learning as GAIL?
*What's the complexity of implementing the proposed method compared with the baselines? Can we easily use it as a better alternative for HER in goal-conditioned policy learning and GAIL in imitation learning without much additional cost of tuning or running the algorithm?

***Post Rebuttal***
Thank the authors for the response. My main concern about the generality of the proposed method is not fully addressed. The baseline HER and GAIL can easily be applied in the domains with discrete action space. Also, the base RL approach for discrete action space exists. Experiments will be more convincing if the proposed method can outperform the baselines on various domains.

---

> ### Author Response · Authors · 2020-11-16
> **Rebuttal**
>
> Thank you for your review. We hope to address some of your concerns in the following:
>
> Regarding the example in Figure 1, yes the figures show the policies learned by HER and UVD. Note, however, that the tabular implementation looks fairly different from one using function-approximation so learning curves are not going to be particularly meaningful. Instead, we rely on intuition to convince the reader that the result must be true. Both methods learn a value-table for each goal. In the tabular case, density estimation simply means counting and UVD is counting how much (discounted) time the agent spends at the goal compared to other states. It should be immediately apparent (and is shown in the paper) that this is equivalent to the true value-function, so it is no surprise that UVD learns the correct policy. Meanwhile for HER, the value-table for the true-goal only gets updated if the true-goal has  been achieved. This means the value of the pit in that table can never change because you cannot escape it. It is then clear that HER cannot learn the correct policy if it cannot estimate the value of the pit.
>
> Regarding additional hyper-parameters introduced by RealNVP and how easy they are to tune: the density estimator does need to be tuned as it can overfit to the collected experience or lack precision. Comparing this with the baselines, GAIL is based on adversarial training which can be brittle and challenging to implement well. HER is a much simpler approach that does not require much tuning; however, as we show it can also simply be wrong. We would recommend to try HER first due to its simplicity and since it works well empirically, but believe that there is nevertheless a need for methods that actually behave correctly such as UVD.
>
> Regarding how RealNVP is applied, normalizing flows like RealNVP allow us to use regular maximum-likelihood with tractable gradients to train a density estimator as long as we have samples from the distribution. In this case, the samples we need are given by the roll-outs.
>
> Regarding the time-horizon, to train the density estimator, we collect triples of state, action and a future state based on a geometric distribution. We do indeed limit how far apart state and future state can be as the variance of the training samples would otherwise be incredibly large and it becomes difficult to train the density estimator. This is mentioned in the main body of the paper but further discussed in Appendix B.1. In practice, we set the value to 4 and didn’t tune it further. This means that the density estimator learns a good value approximately 4 steps away from the goal. The randomly sampled goals for the TD-update thus have to be within 4 steps of the actual goal to get a first learning signal, which defines a sufficiently large region. Higher values trade off the size of this region with variance in the training set of the density-estimator.
>
> Regarding image-based state-spaces, RealNVP as a density estimator has been introduced for use on images and should therefore perform well or even better (due to tuning) in such domains. The remainder of UVD is straight-forward RL and should work well in such spaces as well; however, note that image-based flows tend to be very large and slow models so the computational needs would  be much higher. We would expect a single training run to require several days to train which is why we did not attempt it.
>
> Regarding discrete action-spaces, the general principle of our method is equally applicable to discrete action-spaces and suitable density-estimators exist; however, in our implementation we use TD3 as the underlying RL approach which assumes a continuous action space.

---

### Official Review · AnonReviewer3 · 2020-10-28
**Results Lacking.**

**Rating:** 4
**Confidence:** 4

**Review:**

This paper proposes a new method to compensate for "hindsight bias". This bias can occur in stochastic environments, during the hindsight relabeling process, when the value of actions are overestimated due to the probability of the success of that action in that state being assumed deterministic. The paper proposes Universal Value Density Estimation to account for this overestimation. The method learns approximates the proper goal conditioned Q function using a value density estimate.

Well the discussion at the end of the related work section will most related papers is interesting it's a bit difficult to start to see the novelty and addition this paper has to this problem when previous methods have solved this task for finite Horizons.

The example in figure 1 is very odd it shows an action choice that is basically impossible to cross so you would never end up constructing or relabeling a trajectory that would reach another state beyond the black hole where the agent would fall off the cliff.

Algorithm 2 does not appear to be referenced in the paper, It would be far more helpful if the authors could highlight the differences between these algorithms carefully in the text.

The results in this paper we only shown for a single environment. This does not elicit confidence in the paper's findings. Do either of the domains chosen for the goal conditioned rl tasks have stochastic transitions that would lead to a need for UVD? Artificially adding noise does not seem like a motivating situation for the use of the method...

---

> ### Author Response · Authors · 2020-11-16
> **Rebuttal**
>
> Thank you for your review. We believe your concerns boil down to four points, which we hope to address in the following.
>
> Regarding novelty, this may be a misunderstanding based on our characterization of TD-models, but to the best of our knowledge no previous method solves the problem of hindsight bias for finite horizons. Our approach which characterizes the goal-conditioned value-function as a density function and uses density estimation to estimate it is entirely novel, as is its application to imitation learning.
>
> Regarding Algorithm 2, it is referenced at the end of the section on Value Density Imitation Learning. As is clearly laid out in the abstract as well as the introduction, we propose two approaches: one for goal-conditioned RL, one for imitation learning. Each approach has its own section in the paper and a corresponding algorithm box.
>
> Regarding the example in Figure 1, this is a variation of the seminal cliffwalk domain where noisy transitions are a consequence of the transition dynamics rather than exploration. We show that transition noise in this domain means that HER cannot possibly learn the correct policy. Such terminal failure states are incredibly common (think of a robot colliding with an object), but they are not necessary for hindsight bias to occur. In domains without it, this bias is still present, but potentially less severe and therefore less illustrative.
>
> Regarding the choice of domains, besides the gridworld example, we evaluate UVD and 2 baselines on 4 tasks and VDI and one baseline on 2 tasks with 3 to 4 different settings across different seeds. While it would always be good to add more environments, we are limited by our computational resources. Regarding more naturally occurring noise, any simulator will only have artificially added noise so it is unclear how such an environment could be built. A real robot with naturally occurring noise would be a great motivating example but it is also clearly beyond the scope of the paper. We chose our domains to show that our approach can perform well on standard benchmark tasks and that even simple modifications can break the current state-of-the-art on this benchmark. If Gaussian noise breaks your method, you can assume it to break in real world tasks as well.

---

> > ### Comment · AnonReviewer3 · 2020-11-21
> > **Reponse**
> >
> > "Regarding novelty" of "hindsight bias for finite horizons"
> > The paper indeed provides a method for this problem, but it is also important to motivate the problem's importance. The paper can convince the reader that this is an important or problem in practice that needs to be addressed.
> >
> > "choice of domains"
> > This issue is related to the above comment. While a few environments are used in the paper, they are limited in number and application. It could help to try other forms of noise to understand the robustness of the paper's method, potentially similar to the cliff domain. Running the method on a robot in the real world may be out of the paper's scope; however, the paper claims that the method will help in the real world.

---

### Official Review · AnonReviewer1 · 2020-10-29
**Updated review for UVD/VDI**

**Rating:** 6
**Confidence:** 4

**Review:**

In this paper, the authors offer the perspective that the goal-conditioned Q-value for state-action $(s,a)$ can be viewed as the density of that goal ($g$) under a predictive model $F(g|s,a)$. The density estimator is modeled with normalizing flows (RealNVPs) and trained with MLE using policy rollouts. The estimator is integrated with TD-learning and actor-critic algorithms for goal-conditioned-RL and imitation-learning are derived. Experiments for the former include a comparison with HER on fetch-push/slide tasks, while for IL, the proposed approach is compared with GAIL on two locomotion tasks and shown to exhibit superior demonstration-efficiency.

I would like to appreciate the authors for an excellent presentation of the material. The notation used is rigorous and this makes understanding the hairy parts of the approach easier. Unifying goal-conditioned-RL and IL under a common view of training an agent to reach desired states and leveraging density estimators to achieve that is an interesting approach. I do have some concerns which (currently) hold me from rating this paper higher:

1.	UVD in stochastic domains: It is argued that HER suffers from hindsight bias in stochastic domains; Figure 2d empirically shows this. However, the dip in the performance of UVD from deterministic to stochastic Fetch-slide is also noticeable. I am wondering if this is because the density estimation becomes challenging in highly stochastic domains, since the support of the modeled distribution (over $g$) gets wider with increasing stochasticity, assuming the time-horizon of the density estimator is the same. Could the authors please comment on this observation from Figure 2?

2.	Adding Fetch-push-noise results in Figure 2, using the same noise-model as that used for Fetch-slide-noise, would make for a better claim.

3.	Locomotion tasks modification: These tasks are modified by removing the x/y velocities from the state given to the agent at each timestep. However, these components are used in the internal transition-dynamics function and also the reward function. Therefore, I reckon that this modification changes the MDP into a POMDP. Does that sound correct? If yes, it’s a bit strange that the RL machinery developed for MDPs (GAIL and VDI) is applied to a POMDP task. For reference, it would be good to include the results w/o the velocity removal in the Appendix.

4.	Algorithm 1 vs. Algorithm 2: Is there a particular reason why Algorithm 2 does not use the immediate rewards and the density-estimator from the "next"-state-action in the TD-error computation, as it does in Algorithm 1?

Typo: $F_\omega$ should be $F_\phi$ in Algorithm 2?


===== POST REBUTTAL UPDATE =====

I maintain my score of 6.  While the proposed approach is interesting, the experimental section could be strengthen to firmly stake a claim that it's broadly applicable.  (c.f. point 2. in the review)

---

> ### Author Response · Authors · 2020-11-16
> **Rebuttal**
>
> Thank you for your review. We hope to address each of your 4 concerns in the following.
>
> Regarding the performance in the stochastic domain, please note that the added stochasticity means that there is always a chance that the robot will flick the disk in the wrong direction so some deterioration is to be expected. We don’t know what is achievable by the optimal policy in this domain and it is certainly possible that there are also some limitations due to the algorithm and some room for improvement.
>
> Regarding fetch-push-noise, we tried this variation but did not perform extensive evaluation on it since hindsight bias does not play a big role here. This is because underestimating the noise means that HER might push the box in the wrong direction, but unlike in fetch-slide, the agent can recover from this mistake afterwards (the box doesn’t go anywhere). Completion time and reward will suffer by a small amount, but this is more difficult to measure with certainty.
>
> Regarding the difference between Algorithm 1 and 2, this is a good catch. We omit the reward because talking about reward in an imitation-learning context tends to be confusing (there is no environment-reward). Note that usually, the state-space is continuous and the reward can be set to 0 in continuous state-spaces (it is only non-zero if you match the demonstration *exactly*. In other words, we can and should choose the epsilon ball around the demonstrations in which reward is given to be arbitrarily small). The other difference is whether sampling the transition to the next state is inside of the max or not. It should indeed be outside of the max as it is in Algorithm 1 to avoid bias in stochastic domains.
>
> Regarding the modified locomotion tasks, we make the argument in the paper that the original locomotion tasks are easily gamed and the results are meaningless for imitation learning. All an algorithm has to do is find the feature with the least variance and use the squared distance on that feature as the reward. Since adversarial approaches train a discriminator as a distance function, they are biased to do exactly that. After making this argument, it would be odd for us to also provide results on this domain. It is true that the modified task is a POMDP, but that is also true of the unmodified tasks since not all velocities are included in the state-space. It is also true of most other RL benchmarks that are commonly used, including Atari. RL algorithms are generally able to handle partial observability in practice. Note that the expert is trained on the same variation of the domain and so does not have any information that the agent doesn’t have.

---

### Decision · Program_Chairs · 2021-01-07
**Final Decision**

**Decision:**

Reject

**Comment:**

This paper makes an interesting connection between the density matching in imitation learning and reaching the goal state in goal-oriented reinforcement learning. Reviewers generally expressed that the paper proposes an interesting approach, but some aspects of the paper need room for improvement. By reinforcing the experiments that address the reviewers’ various concerns, this paper will make a good contribution towards reinforcement learning research.